# Spin-Coated Polysaccharide-Based Multilayered Freestanding Films with Adhesive and Bioactive Moieties

**DOI:** 10.3390/molecules25040840

**Published:** 2020-02-14

**Authors:** Joana Moreira, Ana C. Vale, Ricardo A. Pires, Gabriela Botelho, Rui L. Reis, Natália M. Alves

**Affiliations:** 13Bs Research Group, I3Bs—Research Institute on Biomaterials, Biodegradables and Biomimetics, University of Minho, Headquarters of the European Institute of Excellence on Tissue Engineering and Regenerative Medicine, Avepark, Barco, 4805–017 Guimarães, Portugal; joana.lagoa@hotmail.com (J.M.); rpires@i3bs.uminho.pt (R.A.P.); rgreis@i3bs.uminho.pt (R.L.R.); 2ICVS/3B’s, Associate PT Government Laboratory, 4710-057 Braga/4805–017 Guimarães, Portugal; 3The Discoveries Centre for Regenerative and Precision Medicine, Headquarters at University of Minho, Avepark, Barco, 4805–017 Guimarães, Portugal; 4Department of Chemistry, University of Minho, Campus de Gualtar, 4710–057 Braga, Portugal; gbotelho@quimica.uminho.pt

**Keywords:** layer-by-layer, spin-coating, freestanding films, catechol-modified polymers, nacre-inspired, nanocomposite

## Abstract

Freestanding films based on catechol functionalized chitosan (CHI), hyaluronic acid (HA), and bioglass nanoparticles (BGNPs) were developed by spin-coating layer-by-layer assembly (SA-LbL). The catechol groups of 3,4-dihydroxy-l-phenylalanine (DOPA) present in the marine mussels adhesive proteins (MAPs) are the main factors responsible for their characteristic strong wet adhesion. Then, the produced films were cross-linked with genipin to improve their stability in wet state. Overall, the incorporation of BGNPs resulted in thicker and bioactive films, hydrophilic and rougher surfaces, reduced swelling, higher weight loss, and lower stiffness. The incorporation of catechol groups onto the films showed a significant increase in the films’ adhesion and stiffness, lower swelling, and weight loss. Interestingly, a synergetic effect on the stiffness increase was observed upon the combined incorporation of BGNPs with catechol-modified polymers, given that such films were the stiffest. Regarding the biological assays, the films exhibited no negative effects on cellular viability, adhesion, and proliferation, and the BGNPs seemed to promote higher cellular metabolic activity. These bioactive LbL freestanding films combine enhanced adhesion with improved mechanical properties and could find applications in the biomedical field, such as guided hard tissue regeneration membranes.

## 1. Introduction

Research often finds inspiration in the multifunctional materials present in nature to design new biomaterials. Two examples suitable for the design of freestanding films are the robust nacre structure [1,2] and the water-resistant adhesive proteins secreted by marine mussels adhesive proteins (MAPs) [3].

Nacre is a natural laminate composite, which composes the inner layer of mollusk shells [3], with remarkable high fracture toughness and energy absorption properties [4]. Nacre’s toughening mechanism relies on its unique lamellar and staggered structure with multiple-length sizes, composed of 95 wt. % aragonite and 5 wt. % organic material [5]. In addition, the organic material has an effect of a viscoelastic glue, providing a crack-deflection layer [6]. A different biomimetic approach can be also envisaged based on the extremely strong and multisurface adhesion properties of marine mussels under wet conditions [7,8] to enhance film surface adhesion. Marine mussels secrete adhesive proteins (MAPs) that contain the amino acid 3,4-dihydroxy-l-phenylalanine (DOPA), which has a critical role in the wet adhesion due to the presence of catechol functional groups [9,10]. Catechol groups are able to form strong covalent and noncovalent bonds with various inorganic/organic/metallic surfaces [7,10,11], and even biomacromolecules, such as pig gastric mucin glycoprotein [10,12].

Layer-by-layer (LbL) is one of the most interesting techniques to produce nacre-like structures due to its ability to use different materials and to control the thickness of the produced structure at the nanoscale level [1,13,14]. LbL assembly relies on the sequential adsorption of different materials on a substrate surface driven by multipoint interactions, such as electrostatic contacts, Van der Waals forces, and hydrogen bonding [13,15,16]. The power of this technique relies in the wide choice of molecules available [17]. Among them, the use of natural origin polymers in LbL has experienced an exponential growth, being used for a wide variety of applications due to their well-recognized potential, including biocompatibility, often biodegradability, water solubility, similarity with biological tissues, and cell signaling [16,18,19,20].

The combination of LbL with the spin-coating technique, also known as spin-assembly LbL (SA-LbL), has emerged as a promising alternative for the time-consuming traditional dip-coating technique, where the adsorption steps take a few minutes, while spin-coating takes only a few seconds. Spin-coating is a simple, fast, highly controllable, and reproducible technique that involves the deposition of a solution onto a spinning substrate, resulting in the formation of very homogenous films that can be easily detached from the surface and with smooth surfaces due to the high-speed horizontal polymeric diffusion during the spin-coating process [21,22,23,24]. The technique can easily be automated, which makes it more attractable for scaled-up production. It has been shown that the underlying mechanism for film formation in SA-LbL involves the mechanically induced entanglement between polyelectrolytes (PEs) chains of different layers [24]. Final film thickness and other properties depend on the PEs nature, such as viscosity, charge density, surface tension, and solvent pH, and the parameters chosen for the spin process, namely spin speed, spin time, contact time, and acceleration [21].

Inspired by the layered organic-inorganic structure of nacre and the adhesive properties of MAPs, this work proposes the development of bioactive nanostructured freestanding films with enhanced adhesion and mechanical properties. The freestanding films produced through SA-LbL are composed of the natural polycationic chitosan (CHI), polyanionic hyaluronic acid (HA), and by bioglass nanoparticles (BGNPs). Moreover, both incorporated polymers were modified with catechol groups to improve the films’ stability and their adhesive properties. The incorporation of an inorganic phase has the dual aim of enhancing the mechanical properties and providing bioactivity to the films [11,25]. Additionally, the films were chemically cross-linked with genipin, a naturally iridoid derivative from gardenia fruit (*Gardenia jasminoides* Ellis) [26] to increase their stability in wet state.

The above-mentioned materials selected for this study, with the exception of the modified CHI, have already been successfully used to produce LbL coatings and freestanding membranes [18,27,28,29,30,31,32,33,34,35]. Nevertheless, all the reported studies have used the dip-coating deposition method. Importantly, this is the first report of the production of thick (>1 µm) spin-coated natural-based freestanding films, since most studies have developed spin-coated films with nanometer thickness [21,22,24]. Furthermore, the present study describes the production and characterization of the first LbL freestanding films that incorporates both catechol-modified CHI and HA, combined or not with BGNPs, emphasizing the effect of these organic and inorganic components on their physicochemical, adhesive, bioactive, and biological properties. It is expected that these films could be potentially used as guided hard tissue regeneration (GTR) membranes, due to their combination of biodegradability, bioactivity, and adhesiveness.

## 2. Results

### 2.1. UV-Vis Analysis of Catechol-Modified Polymers

CHI and HA were successfully modified with catechol groups through N-(3 dimethylaminopropyl)-N′-ethylcarbodiimide hydrochloride (EDC) chemistry, confirmed by the UV-Vis absorbance spectrum shown in Figure 1 and also through ^1^H-NMR analysis shown in Figure A1 and Table A1. In particular, a maximum absorbance peak at a wavelength around 280 nm can be seen in Figure 1 for both modified CHI and HA, confirming the presence of the catechol groups [8,36,37,38]. Catechol-conjugated CHI (CHIcat) and catechol-conjugated HA (HAcat) were compared with a standard curve built of each catechol group donor, namely hydrocaffeic acid (HCA) and dopamine hydrochloride (DN), showing a catechol substitution degree of 18% and 35%, respectively.

CHIcat yields a substitution degree of 18%, a similar value to those reported in other studies using EDC chemistry and hydrocaffeic acid [37,38,39], confirming a successful catechol functionalization of CHI. Moreover, when dissolved in 0.15 mol/L NaCl aqueous solution, CHIcat solutions were not transparent as CHI solutions, exhibiting a light pink color. Also, accordingly with Kim and coworkers [37], the conjugation of catechol groups onto CHI backbone enhanced its solubility in a wide pH range (2–7), while unmodified CHI was only soluble under acid conditions. In basic environments, a catechol group loses two electrons and two protons, subsequently becoming an oxidized quinone form that is highly reactive with amine groups or another catechol groups via Michael-type addition reaction [8,27,40]. Definitely, the presence of unoxidized catechol groups is an important factor in the preservation of the mucoadhesive properties of CHIcat [37]. In physiological conditions, the oxidized catechol groups can be detected through the decrease on the absorption at 280 nm and new peaks rising at higher wavelengths (300 and 475 nm). Since new peaks were not detected at wavelengths higher than 300 nm, it can be assumed that the catechol groups were not oxidized.

HA was also successfully modified with catechol groups, holding a substitution degree of 35%, similar to the values found in literature [30,31,32,36,41,42]. Moreover, when dissolved in 0.15 mol/L NaCl aqueous solution, it presented a slightly silver coloration.

### 2.2. Zeta Potential Measurements

The pH and ionic strength influenced the LbL film growth, and the Zeta potential could give the charge of each polyelectrolyte solution, which is a key factor for its stability and, consequently, for the successful LbL assembly and performance [43]. The zeta potential values of the polymers and BGNPs (Table 1) were similar in module at pH 5.5, indicating its feasibility to form an LbL film. Hence, CHI and HA are both weak polyelectrolytes, with charge roughly between −18 and −23 mV, being only partially charged at moderate pH near their pKa [44]. Moreover, assuming that their substitution degrees were not so high, the polymers’ charge after the catechol modification did not change significantly. Once only a fraction of amine groups in CHI or carboxylic acid groups in HA were modified with catechol groups, the polymer preserved its charged nature for use in LbL assembly, while integrating the beneficial adhesive features of catechol groups into LbL films [45]. Regarding the nanoparticles, zeta potential values of 30 mV (either positive or negative) indicated particle stability. The produced BGNPs yielded a zeta potential of −20.50 ± 0.76 mV, indicative of a slight tendency to aggregate [46].

### 2.3. BGNPs’ Morphological and Elemental Characterization

The morphology of the BGNPs observed by scanning electron microscopy (SEM), as well as its energy-dispersive X-ray spectrometry (EDS) quantification, is shown in Figure 2. Spherical-shaped BGNPs with diameter below 100 nm are visible. Nevertheless, the diameter of most of the particles was below 50 nm, as expected from the followed production protocol [47].

EDS analysis was used to quantify the ternary BGNPs’ final ratios in atomic percentage (At. %). The measured values were very close to the initial formulation (SiO_2_:CaO:P_2_O_5_—55:40:5), only presenting slight deviations. The reason for these differences may be the removal of free calcium ions during the washing step that were not incorporated in the silica or phosphoric gel network. Indeed, in the sol stage, almost 100% of Ca dissolved in the pore liquor as calcium nitrate, and it would only deposit in the drying step [27,40]. The BGNPs’ specific surface area, total pore volume, and average pore size, obtained by the BET method, was found to be 125.5 m^2^ g^−1^, 0.148 cm^3^ g^−1^, and 58.507 Å, respectively. These values were higher than those of melt-derived bioglass [48] and higher than some sol-gel derived BGNPs with similar formulations [47,49]. Sol-gel derived BGNPs have higher nanoporosity and, therefore, present higher dissolution rates and enhanced bioactivity [50]. These results confirmed the successful production of SiO_2_-CaO-P_2_O_5_ BGNPs.

### 2.4. Production of the LbL Freestanding Films

Different LbL freestanding films based on CHI, HA, and BGNPs, and their catechol-modified counterparts, were assembled by SA-LbL. The film formulations are schematically represented in Figure 3 and the spin-coating parameters are indicated in Table 2. Films composed by unmodified polymers and BGNPs were produced with 100 tetralayers and 50 tetralayers. Films composed by catechol-modified polymers and BGNPs were produced with 50 tetralayers. All these films were easily detached from the substrate. Catechol-Containing films with 100 tetralayers were not possible to obtain due to the strong complexation between PEs, resulting in an increased wet film thickness during the spin-assembly, loss of surface tension between the film and the substrate and, ultimately, its projection to the spin-coater’s exhausting bowl.

Positively charged CHI (or CHIcat) was initially adsorbed to the inert hydrophobic polypropylene substrate through Van der Waals interactions followed by the adsorption of anionic HA (or HAcat). The sequential deposition of the different oppositely charged PEs resulted in the formation of interdigitated layers bound by electrostatic interactions and polymer entanglements [24]. BGNPs were incorporated within the multilayers by electrostatic interactions due to their negative charge, as confirmed by the zeta potential measurements, and also by Van der Waals interactions. The freestanding films were easily detached from the polypropylene substrate due to their hydrophobicity and weak interactions between the initial CHI layer and the substrate. At the macroscopic scale, the obtained films were homogeneous, had no deformities, and could be easily manipulated with tweezers. However, BG100 and BG50 tended to form cracks during the drying process. Before cross-linking, the polymeric-only films were transparent and, when comprising catechol-modified polymers (Cat50 and CatBG50), displayed a yellowish tone. The films incorporating BGNPs presented a light-grey color, which is characteristic of the particles, indicating a homogeneous dispersion and a good interfacial bonding and interaction between the BGNPs and the PE layers.

After cross-linking with genipin, the films had a blue/green coloration (Figure 4). While the dry films without BGNPs were flexible, BGNPs-containing BGNPs-containing films were brittle. Nevertheless, all the films were flexible in the hydrated state. The films were cross-linked to improve their stability when exposed to aqueous media. The use of genipin as cross-linker allowed full exploitation of the mucoadhesion properties of catechol [38], since it only consumes amino groups from chitosan chains for cross-linking, as illustrated in Figure 3.

### 2.5. Films’ Morphological and Topographic Characterization

The morphology at the micrometer level of both surfaces of the LbL films and their cross-section were imaged by SEM (Figure 5). To better understand the films’ topography, AFM imaging was also performed (Figure 6) and allowed the calculation of the root mean squared (RMS) roughness (Figure 7). For analysis purposes, “bottom surface” refers to the surface facing the substrate and “upper surface” to the last produced layer.

SEM images revealed that the surface of all films was homogeneous, flat, and with no defects, which revealed the efficacy of the film detachment technique. BGNPs containing films displayed rougher surfaces due to the inclusion of BGNPs in the polymeric matrix, where the heterogeneous formation of particle aggregates may have also influenced such topography, as reported in other studies [28,29,30,51]. BG50 presented the highest RMS value, which, in comparison with BG100, suggests that the increase of the layers number promotes a more uniform and homogeneous BGNPs deposition and decreases the aggregates sizes, as reported by Couto and coworkers [29]. On the other hand, the polymeric films presented a smoother surface, which was confirmed by their lower RMS roughness.

The upper surfaces tended to have higher RMS roughness compared to the bottom ones, which is in accordance with other studies [52]. In fact, the increased roughness can be explained by the significant presence of polymeric loops and tails in the upper side of the films, producing rough surface areas, as visible by AFM. In addition, due to the substrate’s flat topography, the bottom surface was smother [52,53]. Interestingly, the substrate pattern effect (shown in Figure A2) can be particularly noticed on the polymeric-only films with the bottom surfaces displaying similar RMS to the substrate RMS (CTR100 25.42 ± 4.92, and, CTR50 14.50 ± 3.96, and Cat50 25.85 ± 1.86 nm, substrate 21.76 ± 4.47 nm). The same was not observed for BGNPs containing films. Moreover, there were no significant differences in the RMS values determined for films with and without catechol-modified polymers, which is in agreement with the work of Rego and coworkers [30].

The thickness of the freestanding LbL films was determined using the cross-section SEM images (Table 3). BGNPs-containing films displayed higher film thickness compared to the polymeric films, mostly due to the size contribution of the nanoparticles (<50 nm diameter) at each tetralayer. Film thickness was also measured before cross-linking, the cross-sectional SEM images are shown in Figure A3. A slight thickness increase was observed upon cross-linking the films, with the exception of CTR100, which was slightly inferior. The results are in accordance with other works, where cross-linking increased the stiffness of the multilayers without significant changes in PEMs’ thickness in the dry state [52].

### 2.6. Thermogravimetric Analysis (TGA)

The thermal stability and the weight composition of the LbL films were evaluated by thermogravimetric analysis (TGA). This was valuable information that confirmed in which temperature the membranes started to degrade, which is critical to optimize the production and further processing of the studied membranes. Figure 8 illustrates the obtained results, where three different slopes can be identified. The first, until 200 °C, was mostly related to the release of strongly adsorbed water. The second, between 200 °C and 400 °C, where the major weight loss was observed, was due to the decomposition of the polymeric materials. Above 600 °C, the weight loss tended to stabilize, leaving a residual weight of carbonaceous material from the polymers’ carbon backbone and from the BGNPs. Contrarily, above 700 °C, CTR50 and Cat50 presented a decrease in weight loss due to the loss of the residual ashes.

The total residual weight of each sample measured after thermal decomposition at 770 °C was: CTR100, 34.0%; CTR50, 35.6%; BG100, 67.3%; BG50, 57.2%; Cat50, 35.9%; and CatBG50, 58.7%. Higher residual weight was observed for BGNPs containing films, confirming the successful incorporation of nanoparticles within the PE layers. The approximate BGNPs’ weight content of each film was: BG100, 50.5%; BG50, 33.5%; and CatBG50, 35.5%. These results show that the increase of the layer number led to a higher BGNPs’ incorporation. Moreover, CatBG50 presented slightly increased BGNPs’ weight content compared to BG50, due to the presence of catechol groups in CatBG50, which were able to create strong bonds with several materials, including glass [12,27,54]. Rodrigues and coworkers [28] obtained 12% higher BGNPs’ incorporation in dip-coated LbL films comprising HAcat, CHI, and BGNPs than in its control. This discrepancy is possibly related to the films’ assembly technique. In the dip-coating process, the substrate was dipped for 20 min in the BGNPs’ suspension, allowing sufficient time for the BGNPs to establish electrostatic interactions with the pre-deposited layers, while the BGNPs suspension in the present spin-assembly was in contact with the pre-deposited layers without spinning only for 5–7 s. Thus, increased BGNPs’ incorporation may be possibly achieved by increasing the contact time of the BGNPs’ suspension with the pre-deposited layers.

### 2.7. Water Contact Angle (WCA) Measurements

The results of the WCA measurements of the produced LbL films are depicted in Figure 9. There was no difference in wettability with increasing layer number, nor with the use of catechol-modified polymers, as also reported by Neto and coworkers [31]. For all film formulations (both cross-linked and uncross-linked), the bottom surface presented a more hydrophobic behavior when compared to the upper one, which is in accordance with the findings reported by Rodrigues and coworkers [28]. Such differences in wettability could be used to produce films with asymmetrical biologic response. This can be especially relevant in GTR applications, as the two surfaces of the films will face the gingival epithelium or the alveolar bone defect. In theory, this could lead to an improved bone regeneration, while diminishing the risk of invasion of epithelium cells to the defect [55,56,57].

### 2.8. Swelling Behavior and Degradation Studies

Hydration is a critical parameter in LbL films, impacting film thickness, mechanical/viscoelastic properties, swelling ability, transport properties, and molecular mobility [31]. The swelling kinetics of the cross-linked LbL films at 37 °C in PBS solution is shown in Figure 10. Swelling equilibrium was achieved within 3 h for all the films, with the exception of CTR100 that reached equilibrium after 7 h of immersion. Compared to the polymeric films, BGNPs containing films showed the lowest water uptake and reached equilibrium faster. The lower polymeric content on those films resulted in reduced swelling behavior. Similar findings have been reported for other LbL natural-origin polymer films incorporating BGNPs [28]. In contrast, the polymeric films CTR100 and CTR50 displayed the highest swelling behavior, being higher for CTR100 due to the higher layer number. The same behavior was not observed for BG100 and BG50, where BG100 WU was the lowest, due to the higher amount of BGNPs. The catechol incorporation onto the films Cat50 and CatBG50 led to a water uptake decrease of approximately 45% and 31%, compared to the corresponding controls CTR50 and BG50, respectively. The decrease on the swelling behavior upon the incorporation of catechol moieties has already been reported in literature [41,58,59].

CTR100 and CTR50 present higher swelling ability, typical of natural-based LbL films, which consequently results in thicker films in wet conditions [59,60,61,62]. The reduced charge density related to the use of weak PEs in the film assembly has been associated to the fabrication of thicker LbL films and higher changes in thickness upon hydration when compared to other film combinations, i.e., strong-strong or weak-strong. There is a substantial amount of carboxylic acids with H-bonded carbonyl groups present on CHI-HA films, suggesting a relatively weak charge density on HA as well. Therefore, there was reduced ion pairing between the two PEs, which allowed films to swell significantly when wet [63].

Degradation studies of the films were conducted (Figure A4) to analyze the non-enzymatic hydrolysis and enzyme-catalyzed hydrolysis, which are two common degradation processes in natural polymers [64,65]. For the enzyme-catalyzed hydrolysis evaluation, a lysozyme solution was used, which is the main enzyme involved in the degradation of chitin derivatives, such as CHI, and can be found in human serum in concentrations around 7–13 mg L^−1^ [64].

### 2.9. Dynamic Mechanical Analysis (DMA)

The mechanical and viscoelastic properties of the freestanding films were evaluated by DMA. The storage modulus (E′) and the loss factor (E″/E′) as a function of the tensile loading frequency is presented in Figure 11. The results evidence an increase in E′ with the frequency increase for all the LbL films studied (with the exception of CTR50), a particular behavior of viscoelastic materials [66], being in agreement with other works [52,67]. Such increase was more pronounced for CatBG50, which also reached the highest E′ values (73.13 ± 5.6 MPa), suggesting that the synergetic effect of the catechol groups and the BGNPs’ incorporation resulted in an increased film stiffness, reaching values 6.8 and 1.7 times higher than BG50 and Cat50, respectively. The inclusion of catechol groups revealed to be crucial in increasing films’ stiffness, given that Cat50 reached E’ values 1.9 times higher than CTR50 and comparable to CTR100.

### 2.10. Adhesive Properties

The assessment of the surface’s dry adhesion by atomic force microscopy (AFM) is based on the force exerted on the AFM probe as it was retracted from the surface after being in contact [68]. The interaction between the probe and the substrate was monitored by sensing the deflection of the cantilever. Knowledge of the spring constant allowed a conversion from cantilever displacement into force [69]. In some surfaces, due to binding forces along the contact region, the probe may not separate from the surface at the same point where contact was originally established. Rather, a substantially larger negative load is required before an abrupt separation occurs, known as “pull-off” force or adhesion force [69]. Thereby, this force was determined from the maximum of the adhesion value (in modulus) upon retraction of the surfaces, from a force curve taken at each array [70], as illustrated in Figure A5.

The adhesive strength of the produced films’ adhesion was studied using the AFM operating in the tapping mode (dynamic mode), where the adhesion force was determined through the force-distance (F-D) curves, which is a technique that has already been used to study the single-molecule adhesion force of DOPA [12]. It should be mentioned that the adhesive force herein evaluated can be considered as a “one-side” adhesive system, being mainly dependent on the surface interaction between probe and sample surfaces, and, therefore, it varied with the probe specifications (tip and cantilever) being used. Thus, the AFM characterization was performed with a probe moderately soft with a midrange resonance frequency for consistent high-resolution imaging, and mainly to assess the adhesion enhancement upon using catechol-modified polymers.

Figure 12 shows the mean adhesion force values ± standard deviation for 100 F-D curves from different regions for each film formulation. Figure A6 shows representative histograms of the adhesion force constructed from repeated curves across the surface of each film formulation.

### 2.11. In Vitro Bioactivity Studies

The in vitro bioactivity of the constructed LbL films was analyzed by the capability for apatite formation on the film surface after their immersion in SBF for 14 days. Figure 13 shows the SEM images of all film formulation surfaces, where the formation of apatite-like layers on the surface of BGNPs containing films was observed. The precipitate exhibited the typical cauliflower morphology, containing needle-like nanometric structures, characteristic of bone-like apatite. The EDS spectra (Figure 14a) revealed the presence of higher amounts of Ca and P upon SBF immersion on the BGNPs containing films, while the Si peaks presented before immersion disappeared. Accordingly with the peaks’ evolution mechanism explained by Luz and Mano [47], the EDS spectra revealed decreasing Si peaks and increasing Ca and P peaks during SBF immersion due to the BGNPs’ dissolution, which provided the formation of a Ca-P layer. The presence of carbon and oxygen was also found, the only elements found in the polymeric films’ EDS spectra shown in Figure A7, consistent with an organic material.

XRD experiments were performed to have a deeper insight on the nature of the formed calcium phosphate layer (Figure 14b). The XRD spectrum of all films before SBF immersion essentially exhibited an amorphous profile. After 14 days of immersion, the XRD data revealed a series of diffraction peaks associated with apatite (Figure 14b), i.e., the XRD patterns had the typical hydroxyapatite diffractogram, as previously reported in other works [29,30]. So, it can be concluded that XRD, SEM, and EDS results confirmed the bioactive behavior of the BGNPs containing films, essential for the formation of new bone, which is of utmost importance for their potential application as GTR membranes [55,71].

### 2.12. Cellular Assays

The cytocompatibility of the CTR50, Cat50, and CatBG50 was evaluated by direct contact with SaOs-2 cells cultured on these films during 1, 3, and 7 days. TCPS was used as a positive control, where cells are supposed to have a great proliferation.

Cellular viability was evaluated by performing the MTS assay, where the metabolic activity of the cells can be achieved through the chemical reduction of the MTS compound to formazan. Overall, all conditions presented an increase in cell viability over the time (Figure 15a). Significant differences were only found at day 3, between the TCPS and the polymeric films, CTR50 and Cat50 (*p* < 0.001), and between the TCPS and CatBG50 (*p* < 0.01). At day 7, TCPS and CatBG50 presented the highest cellular viability, being very similar to each other. The significant differences of CatBG50 with the polymeric films revealed that the BGNPs’ inclusion allowed cells to express better metabolic activity.

The cell proliferation over a period of 7 days was assessed by DNA quantification. Figure 15b shows that, along with the culture time, all formulations led to an increase in cell proliferation. Unsurprisingly, there were no significant differences on the first day. At day 3, similar DNA content was obtained from CTR50 and TCPS. At day 7, both CTR50 and CatBG50 presented similar cell proliferation, while Cat50 presented lower values. Significant difference was found between CTR50 and Cat50 (*p* < 0.01). As expected, TCPS presented higher cell proliferation after 7 days.

## 3. Discussion

As it was hypothesized, catechol-containing films led to the construction of thinner films. CatBG50 were thinner than BG50, suggesting that the combination of catechol-modified polymers with BGNPs may be responsible for more compact LbL films, accordingly with other studies [30,31]. The cross-section of the BGNPs-containing films showed some porosity, which was much reduced in the case of CatBG50 compared to BG100 and BG50, revealing the more compact structure of CatBG50 films. Nevertheless, such porosity could be beneficial for biomedical applications, for example, to enhance nutrient transport to cells or to increase the surface area.

Although both films with 100 tetralayers were slightly thicker than their 50-tetralayer counterparts, increasing the layer number did not had a significant impact on film thickness, meaning that the adsorption rate decreased after a certain layer number. This is a known behavior of LbL films, which generally exhibit an exponential growth only in the initial deposition cycles, followed by a linear growth of thickness. According to Michel and coworkers [72], when the film thickness becomes important and when the contact time between the film and the solution is not increased accordingly, there is not enough time for the polycations close to the film/substrate interface to diffuse towards the film/solution interface where inter-polyelectrolyte complexation and film growth occur. Hence, at a certain level of film growth, only a constant part of the film should be affected by diffusion of free chains out of the film, leading again to a linear growth. Lastly, comparing the thickness of spin-coated films with similar formulations of dip-coated ones, it was found that the spin-coated are thinner [33].

The presence of BGNPs revealed to deeply affect the wettability of the films. These results are in agreement with other studies, where the presence of BGNPs at the surface of a composite is able to increase its hydrophilicity [28,73]. Moreover, similar WCA values were obtained for all BGNPs-containing films, meaning that the differences in BGNPs’ content did not have impact on the films’ wettability. The cross-linking of the LbL films decreased the WCA of both surfaces, becoming more hydrophilic, as also observed for CHI-ALG films with 100-layer pairs cross-linked with genipin [52]. The decrease has been related to the increase of cross-linker conjugates within the films, which are richer in hydrophilic groups, such as amine, amides, and hydroxyl groups [74,75,76]. However, the upper surface of BG100 was an exception, as the WCA increased with the cross-linking. Its uncross-linked upper surface displayed the most hydrophilic behavior due to the BGNPs’ incorporation, and consequently its porosity. The cross-linking may have reduced the spaces inside the BG100 films’ structure and between the BGNPs. The same did not happen for CatBG50, probably due to its more compact structure and lower BGNPs’ weight content.

Concerning the stability of the developed films, independently of the used solution, the degradation was more pronounced in the first seven days for all the film formulations, with the exception of CatBG50, which revealed to degrade at a more constant rate. BG100 presented the higher weight loss, due to the higher amount of BGNPs that could be released and dissolved from the films, which is in agreement with previous studies [28,67]. A similar weight loss was observed for CTR100, CTR50, and BG50 in both solutions. A lower weight loss was observed for the catechol-containing films, being CatBG50 the lowest, suggesting that the catechol incorporation increased the film stability. This finding may be related to their lower swelling ability and more compact structure, which could have reduced the quantities of PBS and lysozyme entering into the polymeric matrix. Xu and coworkers also observed that CHIcat hydrogels lost 14–17% of their weight after 3 days immersed in PBS, while CHI hydrogels lost 24% [38].

There are very few reports on the development of LbL freestanding films based on catechol-modified natural polymers [28,33,34,35]. Still, these polymers have been mostly exploited as hydrogel systems, and the catechol presence revealed to increase their overall mechanical properties [36,37,38,77]. Similarly, in the present study, the inclusion of catechol groups resulted in increased film stiffness. Remarkably, when BGNPs were also combined, the highest E′ values were achieved. However, for BGNPs-containing films without catechol-modified polymers, namely BG100 and BG50, lower E′ values were obtained. Probably, without the presence of the catechol groups which act as a “glue”, the BGNPs’ fraction present in these formulations may have been too high and disrupted the film structure. The different behavior of CTR50, i.e., the decrease of E′ with the frequency, could be related to the temperature increase of the film as the frequency increased, thus increasing the molecular frictions and the dissipated energy. Consequently, both storage and loss modulus decreased [78].

Unsurprisingly, higher adhesion values were shown for both films containing catechol-modified polymers, Cat50 and CatBG50. In particular, Cat50 showed to be the most adhesive film formulation, due to the highest content of catechol groups. Thus, by comparing these film formulations with their controls, CTR50 and BG50, the adhesion force was greatly improved. In fact, adhesion enhancement with the presence of catechol groups was also reported for “both-side” adhesive system with LbL coatings comprising CHI and HAcat [31] and including BGNPs [30,32] and also with CHI-HAcat LbL films [33]. Neto and c-workers reported higher adhesive strength on CHI-HAcat coatings due to the catechol modification [31]. Comparing CTR100 and CTR50, increasing the number of layers led to a slight adhesion decrease, accordingly with Fujie and colleagues who reported an adhesion decrease with increasing thickness in films composing CHI and sodium alginate [79,80,81]. Moreover, both BG100 and BG50 present higher adhesion when compared to CTR100 and CTR50, respectively. Contrarily, some previous works from our group [28,67] reported that the adhesion decreased when freestanding films incorporated bioactive nanoparticles. As it was previously mentioned, the adhesive properties herein evaluated consisted of a “one-side” adhesive system tested in an atomic scale, which is different from the “both-side” adhesive system tested by lap-shear tensile tests reported in our previous studies [28,67]. Thus, the different results may be related to the hydrophilic behavior of these films. Hydrophilic surfaces are covered with a nanometer water layer in ambient conditions. If the tip is also hydrophilic, the water layers can join when the tip and sample are close together, forming a capillary neck between them, resulting in higher adhesion forces [82]. Although the used probe was made of silicon, which is hydrophobic, and also silicon forms “native oxide” at room temperature when exposed to air (or water) enough to make its surface hydrophilic [83]. Considering the potential application of these films as GTR membranes, these results are quite promising as their adhesive nature could help the implantation of the membrane in situ.

The results of the preliminary biological assays indicated that the use of either catechol-modified polymers or BGNPs by itself did not affect negatively cell viability. So, one may conclude that these LbL films are noncytotoxic, assuring the materials’ biocompatibility. Contrarily to other findings, there were no significant differences by the incorporation of only catechol-modified polymers (Cat50). Some studies have reported CHI-HAcat coatings with improved cell attachment when compared to CHI-HA [30,31,32]. A previous work [30] showed CHI-HAcat-CHI-BG coatings with improved cell viability and proliferation. Moreover, enhanced cell viability and proliferation was observed when the last layer was HAcat, instead of BGNPs, and similar findings were obtained by using silver-doped BGNPs [32,67].

On the other hand, the combination of BG with catechol groups in CatBG50 led to a higher metabolic activity compared to the other tested film formulation, adhesion, and proliferation similar to CTR50. Such results may be also related to the hydrophilicity of CatBG50, once cells adhere better to moderate hydrophilic surfaces rather than hydrophobic or superhydrophilic surfaces [42,43]. Furthermore, the effect of BGNPs’ ion leachables on promoting metabolic activity is well known. CHI/BGNPs membranes produced through solvent casting showed improved bioactivity compared to CHI-only membranes and promoted cell metabolic activity and in vivo mineralization [28]. Rego and coworkers developed LbL coatings comprising CHI, HAcat, and BGNPs, which displayed lower cell viability compared to the control one at days 3 and 7, possibly related to the excess of BG dissolution that could have increased the pH of the culture medium [30].

Facing these results, CatBG50 presented the advantage of a lower weight loss rate, avoiding this problem while maintaining its bioactive character. Moreover, the stiffness and roughness increase and water uptake decrease also contributed to an enhanced cellular behavior, since cells tend to prefer stiffer and lower hydration surfaces [52,76,84].

## 4. Materials and Methods

### 4.1. Materials

CHI (medium molecular weight) with a degree of N-deacetylation (DD) ranging from 75 to 85% and a viscosity of 200–800 cP, HA sodium salt from *Streptococcus equi* (Mw = 1.5–1.8 × 10^6^ Da), dopamine hydrochloride (DN), N-(3 Dimethylaminopropyl)-N′-ethylcarbodiimide hydrochloride (EDC), hydrocaffeic acid (HCA), and genipin were all purchased from Sigma-Aldrich (St. Louis, MO, USA). Prior to being used, the CHI was purified by a recrystallization process.

Citric acid monohydrate (C_6_H_8_O_7_·H_2_O), ammonium phosphate dibasic ((NH_4_)_2_HPO_4_), calcium nitrate tetrahydrate 99% (Ca(NO_3_)_2_·4H_2_O), ethanol absolute, sodium sulphate (Na_2_SO_4_) and ammonium hydroxide solution (ammonium hydrogen phosphate 98%, maximum of 33% NH_3_), Tris(hydroxymethyl)-aminomethane (Tris buffer), magnesium chloride hexahydrate (MgCl_2_·6H_2_O), and hydrochloric acid (HCl) were all purchase from Sigma-Aldrich. Tetraethyl orthosilicate (TEOS) 99.90%, di-Potassium hydrogen phosphate trihydrate (K_2_HPO_4_·3H_2_O), calcium chloride (CaCl_2_), potassium chloride (KCl), sodium hydrogen carbonate (NaHCO_3_), and sodium carbonate (Na_2_CO_3_) were purchased from Merck Chemicals (Darmstadt, Germany). Sodium chloride (NaCl), sodium hydroxide (NaOH), and glacial acetic acid were purchased from Panreac Química SLU (Barcelona, Spain).

### 4.2. Synthesis of Catechol-Functionalized Chitosan

Catechol moieties were conjugated onto the CHI backbone via EDC chemistry (Figure A8), following an optimized protocol based on previous works [37,38,39]. Briefly, 100 mL of 1% (*w*/*v*) CHI solution was prepared in 1% HCl (5.5 M) overnight. Then, 0.293 g of HCA was dissolved in 5 mL OW and 0.712 g of EDC was dissolved in a mixture of osmotized water (OW) and ethanol (1:1, *v*/*v*). The HCA and the EDC solutions one were added to the CHI solution under intensive stirring, followed by the addition of 1 mol/L NaOH to reach a final pH of 4.8. After 18 h of reaction under nitrogen atmosphere and protected from the light, the product was purified by dialysis for 3 days at 4 °C, using a dialysis membrane tube against acidic water (pH 5.0, HCl solution), and in OW for 1 day, where the solvent was changed every 3–6 h. The final product was freeze dried for 4 days and stored at −20 °C.

### 4.3. Synthesis of Catechol-Functionalized Hyaluronic Acid

Catechol-conjugated hyaluronic acid (HAcat) was synthesized using EDC chemistry (Figure A9) following the experimental procedure proposed by Lee and coworkers [77] with slight modifications. Briefly, 1 g of HA was dissolved in 100 mL of 0.01 mol/L PBS solution, prepared by dissolving one PBS tablet in 200 mL of OW, and the pH was adjusted to 5.5 with 0.5 mol/L HCl or 0.5 mol/L NaOH aqueous solutions. The solution was purged with nitrogen overnight to limit the oxygen interaction with the solution. Then, 338 mg EDC and 474 mg of DN was added to the HA solution with an acidic pH of 5.5 at 4 °C, for 2 h or until reagents’ complete dissolution. Unreacted chemicals and urea by-products were removed by dialysis using OW for a week, followed by 4 days of lyophilization. The produced conjugate was stored in the dark at −20 °C.

### 4.4. Production of Ternary Bioglass Nanoparticles

BGNPs, with the composition SiO_2_:CaO:P_2_O_5_ (mol%) = 55:40:5, were produced using sol-gel chemistry, following the protocol presented in a previous work [47]. TEOS was used as the Si precursor, ammonium phosphate dibasic as the P precursor, calcium nitrate tetrahydrate as the Ca precursor, citric acid monohydrate to promote hydrolysis, ethanol absolute, and ammonium hydroxide (maximum of 33% NH_3_) as the jellifying agent.

Briefly, at room temperature, 7.639 g of calcium nitrate tetrahydrate was dissolved in 120 mL of OW. Then, 60 mL of ethanol and 9.8353 mL of tetraethyl orthosilicate were added to the calcium nitrate solution. The pH was adjusted to 2 with a citric acid solution (30 mL, 10%), and the solution was stirred for 3 h. A second solution was prepared with 1.078 g of ammonium phosphate dibasic dissolved in OW, using an ammonium hydroxide solution to adjust the pH to 11.5. Then, the first solution was added drop-by-drop to the second. During the dripping process, the pH value of solution was kept at 11.5 using an ammonium hydroxide solution. After being stirred for 48 h and aged for 24 h, the precipitate was washed three times with OW, freeze-dried for 7 days, followed by calcination at 700 °C in a muffle furnace for 3 h, finally, a white powder of BGNPs was obtained.

### 4.5. Freestanding LbL Films through Spin-Coating

LbL freestanding films were produced through the SA-LbL onto a circular hydrophobic polypropylene substrate (Ø = 5 cm). CHI and CHIcat were used as the positively charged materials, and HA, HAcat, and BGNPs were chosen as the negatively charged materials. Six different multilayered formulations of films were prepared, as illustrated in Figure 3 and summarized in Table 2, along with the used spin-coating parameters. Prior to the films’ production, the substrate was washed several times with OW and dried under a gentle stream of nitrogen. Solutions of CHI (4 mg mL^−1^, 2% (*v*/*v*) acetic acid, 0.15 mol/L NaCl), CHIcat (4 mg mL^−1^, 0.15 mol/L NaCl), HA (1.5 mg mL^−1^, 0.15 mol/L NaCl), and HAcat (1.5 mg mL^−1^, 0.15 mol/L NaCl) were prepared in OW. The PE concentrations were chosen according to their viscosity, as it affects the solvent evaporation rate and, therefore, it should be similar for both materials. The PEs’ viscosity was assessed as a function of the concentration using a Kinexus Rheometer (Malvern Instruments, Malvern, UK). The results are shown in Figure A10. The used washing solution was 0.15 mol/L NaCl prepared in OW. The pH of all solutions was adjusted to 5.5 with 4 mol/L NaOH or 4 mol/L acetic acid in the case of CHI, and with 0.5 mol/L HCl or 0.5 mol/L NaOH for HA, HAcat, CHIcat, and NaCl. BGNPs were suspended in 0.15 mol/L NaCl (pH = 5.5) at a concentration of 2.5 mg mL^−1^ and sonicated for 15 min before its use to avoid particle aggregation.

Each material was deposited and spun separately, accordingly with the multilayer formulation. Briefly, 1.5 mL amount of each PE solution was deposited at the center of the substrate (3 mL on the first tetralayer). Then, after a 5–7-s period waiting to allow its adsorption (contact time), the materials were spun (2000 rpm, 10 s). Between each deposition step, a washing step was performed (2000 rpm, 5 s) with 0.15 mol/L NaCl aqueous solution (pH 5.5). The multilayer process was repeated until the desired number of tetralayers was achieved.

After the SA-LbL process, the film borders (1–2 mm) were carefully scratched out of the substrate to avoid its attachment to the lateral and bottom surfaces of the substrate. After drying at room temperature for 1–3 days, the films were neutralized by immersing in 0.2 mol/L NaOH for 2 h, followed by 3 h hydration in OW. Then, films were cross-linked with genipin (2 mg mL^−1^) dissolved in ethanol/PBS (1:1) at 37 °C for 14 h. The resulting cross-linked freestanding films were washed with ethanol and, then, with ultra-pure milli-Q^®^ water, three times a day, for 2 days, to ensure complete genipin removal. Finally, the films were dried for 1 day at RT.

### 4.6. Determination of the Catechol-Degree of Substitution

The conjugation of catechol functional groups to CHI and to HA was evaluated by UV-Vis spectroscopy (λ = 200–350 nm). The degree of catechol substitution was determined by the colorimetric assay at the maximum absorbance wavelength of the catechol (maximum wavelength, λmax = 280 nm). Standard solutions of HCA and DN were used to generate a standard curve of catechol concentrations to quantify the catechol content of CHIcat and HAcat, respectively. Then, for HAcat and CHIcat, the degree of catechol substitution is given by the Equations (1) and (2), respectively.
(1)$$DS\;\left(\%\right)=\frac{m_{DN}}{m_{HAcat}}\times 100$$
(2)$$DS\;\left(\%\right)=\frac{m_{HCA}}{m_{CHIcat}}\times 100$$

Additionally, ^1^H-NMR analysis of polymers and their respective conjugates was also performed to detect characteristic peaks that can confirm their successful catechol-functionalization.

### 4.7. Zeta Potential

In the present study, the Zeta potential of BGNPs, and of the polyelectrolytes CHI, HA, CHIcat, and HAcat, was measured in a liquid medium with a Zetasizer Nano ZS equipment (Malvern Instruments, Malvern, UK). The Malvern Zetasizer device uses a combination of the electrophoresis and laser Doppler velocimetry techniques to measure the zeta potential. Suspensions with 0.3 mg mL^−1^ of the BGNPs particles were prepared in OW with 0.15 mol/L NaCl (pH = 5.5) and sonicated for 15 min before the measurements to avoid their agglomeration. Polyelectrolyte solutions of 0.5 mg mL^−1^ were prepared in 0.15 mol/L NaCl and the pH was adjusted to 5.5. The measurements were performed in triplicate at 25 °C, setting a minimum of 10 and a maximum of 100 runs.

### 4.8. Morphological, Topographical, and Elemental Characterization of the Films and of the BGNPs

The surface morphology of both sides and cross-sectional view of the freestanding films, and also the element composition of the films and BGNPs, was assessed using the SEM JSM-6010 LV (JEOL, Tokyo, Japan) coupled with EDS (INCAx-Act, PentaFET Precision, Oxford Instruments, Abingdon, UK). For the observation of the cross-section, the films were fractured using nitrogen. Cross-Sectional SEM views were obtained from the LbL films before and after cross-linking with genipin, and the correspondent film thickness was estimated using the software ImageJ. The surface morphology of the BGNPs was characterized by a high-resolution field emission SEM with focused ion beam (AURIGA Compact FIB-SEM, Carl Zeiss, Oberkochen, Germany). Before each analysis, except for EDS, the samples were sputtered with a gold layer, using a sputter coater EM ACE600 (Leica Microsystems, Wetzlar, Germany). The specific surface area of the BGNPs was measured by determining the N_2_-gas adsorption isotherms using a TriStar II 3020 surface area and porosity analyzer (Micromeritics Instrument Corporation, Norcross, GA, USA). The samples were degassed at 130 °C overnight before measurements.

### 4.9. Atomic Force Microscopy

The topography, roughness, and adhesive strength of the dried freestanding films were analyzed through an AFM Dimension Icon equipment (Bruker, Kontich, Belgium) with an air cantilever (ACSTA, AppNano, Mountain View, CA, USA) with a spring constant (k) of 7.8 N m^−1^ and resonance frequency (f) of 150 kHz, and a silicon probe. For topography and roughness, analysis was performed operating in PeakForce Tapping mode (ScanAsyst) with 512 × 512 pixels^2^ at a line rate of 0.5 Hz. Three images were obtained in different regions of each scanned sample (5 × 5 μm^2^), followed by root mean squared roughness (RMS) calculation. The analysis of the images was performed using the JPK SPM Data Processor Software (JPK Instruments, Berlin, Germany). For the adhesion measurements, tapping mode operating mode was used at 1 Hz frequency. Force maps with 15 × 15 μm^2^ (8 × 8 pixels^2^) were obtained and the dry adhesion force of the LbL films was determined from 100 force-distance curves registered at different film surface regions using the JPK SPM Data Processor Software (JPK Instruments, Berlin, Germany).

### 4.10. Thermogravimetric Analysis

TGA measurements were performed to quantify the percentage of BGNPs present in the films. The Q500 TGA equipment (TA Instruments, New Castle, DE, USA) was used to obtain the mass variation between 40 °C and 800 °C at a heating rate of 10 °C min^−1^. The amount of BGNPs present in each formulation was calculated following Equation (3):(3)$$B\left(\%\right)=\frac{Rf-R}{100-R}\times 100$$
where *B* is the amount of BGNPs (%), *R* the residual weight of the control films (CTR100, CTR50, and Cat50), and *Rf* is the residual weight of the BGNPs-containing films (BG100, BG50, and CatBG50).

### 4.11. Swelling Behavior

The freestanding films’ swelling behavior was assessed by soaking previously weighed dry films in PBS at 37 °C for 5 min, 15 min, 30 min, 1 h, 3 h, 5 h, 7 h, 24 h, and 48 h. At each time point, the films were removed from the PBS solution, and the excess of PBS was gently removed with a filter paper. Three samples were used for each film formulation. The water uptake (WU) percentage was quantified using the following Equation (4):(4)$$WU\;(\%)\;=\;\frac{Ww-Wd}{Wd}\times 100$$
where *Ww* is the weight of the swollen films and *Wd* is the weight in dry condition.

### 4.12. Weight Loss

To evaluate the weight loss (WL) of the freestanding LbL films, previously weighed films (2 × 1 cm^2^) were soaked in a PBS solution and in a PBS solution containing lysozyme from chicken egg white (0.013 mg mL^−1^) and sodium azide (0.02% *w*/*v*) for 1, 7, 14, 21, and 28 days, at 37 °C. At each time point, the films were removed from the solution, washed multiple times with ultra-pure Milli-Q^®^ water, dried, and then weighed. Three samples were used for each film formulation. The weight loss quantification was calculated using the following Equation (5):(5)$$WL\;\left(\%\right)\;=\;\frac{Wi-Wf}{Wi}\times 100$$
where *Wi* is the initial weight of the dry sample and *Wf* is the final weight of the dry sample, at each time point.

### 4.13. WCA Measurements

The short-term WCA of both surfaces of the LbL films, i.e., the “bottom” side being the surface initially in contact with the polypropylene substrate and the “top” being the end-layer of freestanding LbL films, was assessed by water contact angle measurements using an OCA15plus goniometer equipment (DataPhysics Instruments, Filderstadt, Germany). For each different surface, 10 measurements of WCA were made at RT immediately after 1 µL drops of OW were dispensed by a motor-driven syringe. The results treatment was performed using the SCA20 software.

### 4.14. Dynamic Mechanical Analysis

The DMA experiments were performed using a TRITEC 2000B equipment (Triton Technology, Keyworth, Nottinghamshire, UK). The films were cut into 2.5 × 0.5 mm^2^ size, and the thickness was determined in three different points of each sample, using a micrometer (Mitutoyo, Kawazaki, Japan). Prior to the DMA testing, the samples were soaked in a PBS solution overnight, to reach the equilibrium. The measurements were carried out at 37 °C and the films were analyzed, immersed in a PBS solution, and placed in a Teflon^®^ reservoir. The films were clamped in the DMA apparatus with a grip distance of 10 mm. After equilibrating at 37 °C, the DMA spectra were obtained, performing a frequency scan from 0.1 to 15 Hz. A static preload of 1 N was applied during the tests to keep the sample straight. Three specimens were tested for each condition.

### 4.15. In Vitro Bioactivity

The bioactive character assessment of the films was exclusively performed in vitro. Samples of each film were immersed in SBF at 37 °C for 14 days. After that time, the samples were carefully washed in ultra-pure Milli-Q^®^ water and dried at RT for 24 h. SBF was produced following the protocol developed by Kokubo and Takadama [85]. The bioactive character of the tested materials was assessed by SEM and EDS with a JSM-6010LV SEM-EDS (JEOL, Tokyo, Japan), and by XRD using Bruker AXS D8 Discover model (Bruker, Kontich, Belgium) operated at 40 kV and 40 mA using Cu Kα radiation. The detector was scanned over a range of 2θ angles from 15° to 60° at a step size of 0.04° and dwell time of 1 s per step. The analysis for phase identification was performed using analytical software EVA. The crystalline phases were indexed using the ICDD-2015 database (International Center for Diffraction Data).

### 4.16. Cellular Assays

#### 4.16.1. Cell Seeding

Attending the guided hard tissue regenerative applications envisaged for the constructed freestanding films, human primary osteosarcoma cell line (SaOs-2) obtained from European Collection of cell cultures (ECA CC, UK) were used, in order to test the in vitro cytotoxicity through direct contact tests with the catechol-modified and control films with 50 layers (Cat50, CatBG50, and CTR50) for 1, 3, and 7 days of cell culture. Cells were cultured in complete Dulbecco’s modified minimum essential medium (DMEM, Sigma-Aldrich, St. Louis, MO, USA) with low glucose and phenol red; supplemented with sodium bicarbonate for cell culture, 10% fetal bovine serum (FBS, Sigma-Aldrich, St. Louis, MO, USA), and 1% antibiotic/antimycotic (Life Technologies™, Paisley, UK); and incubated at 37 °C in a humidified air atmosphere of 5% CO_2_ until their confluence. The medium was replaced every 3 days. At 90% of confluence, cells were washed with PBS and detached by a chemical procedure with TrypLE™ express solution with phenol red (Life Technologies™, Paisley, UK) in order to obtain cells between passage 19 and 21 used for this study. Prior to cell seeding, samples with 5-mm diameter were sterilized by exposure to UV light for 30 min and immersion in 70% (*v*/*v*) ethanol for 30 min. Then they were washed three times with PBS and immersed in DMEM for complete swelling. So, cells were seeded above the surface of the freestanding films by adding a cell suspension in DMEM with a density of 2 × 10^4^ cells per mL. Tissue culture polystyrene (TCPS) was used as positive control. Finally, samples were incubated at 37 °C in a humidified air atmosphere of 5% CO_2_.

#### 4.16.2. MTS Assay

Cell viability was assessed using the MTS colorimetric assay (CellTiter 96^®^ AQueous One Solution Cell Proliferation Assay, Promega, Madison, WI, USA) after 1, 3, and 7 days of culture. At each time point, the culture medium was removed, and the samples were rinsed with sterile PBS. Serum-Free culture medium (DMEM) and without phenol red was mixed with the MTS reagent in a 5:1 ratio and added to each well. All conditions were performed in triplicate and placed in the incubator for 3 h, at 37 °C and 5% CO_2_ atmosphere. The absorbance was measured in triplicate at 490 nm in a new, 96-well plate using a microplate reader (Synergy HT, BioTek, Winooski, VT, EUA) protected from the light. The results were expressed through the obtained absorbance values in each condition normalized by the DNA content as a function of the culture time.

#### 4.16.3. DNA Quantification

The seeding efficiency and proliferation of cells into the developed films after 1, 3, and 7 days of culture was investigated using the PicoGreen dsDNA kit (Life Technologies, Paisley, UK). Initially, cell cultured specimens were washed twice with sterile PBS, followed by cell lysis by osmotic (by adding ultra-pure water) and thermal shock (by freezing the cells at −80 °C for at least 1 h). All conditions were performed in triplicate. The obtained supernatant was analyzed using PicoGreen dsDNA kit (Life Technologies^TM^, Paisley, UK) according to the manufacturer’s protocol. The recovered supernatant was read on a microplate reader (BioTek, Winooski, VT, USA) using a 485 and 528 nm as excitation and emission wavelengths, respectively. Triplicates were performed for each sample and the DNA amount was calculated using a standard curve that relates DNA concentration with fluorescence intensity.

### 4.17. Statistical Analysis

Data were expressed as average ± standard deviation (SD) of at least three replicates. The error bars denote the standard deviation. Normality tests were performed using Shapiro-Wilk tests. Depending if the population was normally distributed or not, parametric or nonparametric were used, respectively. For WCA assays where the population was not normally distributed, Kruskal-Wallis test was employed. For the other characterization tests, the population was normally distributed and ANOVA followed by Tukey’s test were used. The statistical analysis was performed using the GraphPad Prism 7.0 for Windows. Statistical significance was accepted for a (*) *p* < 0.05.

## 5. Conclusions

Nacre-inspired freestanding films were successfully developed by LbL-SA combining catechol-modified natural polymers, CHIcat and HAcat, and BGNPs. CHI and HA were successfully conjugated with catechol groups, with 18% and 35% substitution degrees, respectively. The produced films, with thickness ranging from 6 to 50 µm, presented different properties and bioactive behavior, depending on the overall film composition.

The incorporation of BGNPs resulted in thicker and bioactive films, with rougher and hydrophilic surfaces, reduced swelling, and higher weight loss. Nevertheless, with the exception of the catechol-containing film, CatBG50, BG100, and BG50 were more fragile than the other film formulations, which led to poorer mechanical performances. This might indicate that without the presence of the catechol groups that act as a “glue”, the BGNPs’ fraction present in these formulations may be too high and disrupt the film structure. On the other hand, CatBG50 had half of percentage and held the highest storage elastic modulus determined by DMA. Catechol-containing films, namely Cat50 and CatBG50, revealed enhanced adhesive force and stiffness, lower swelling, and weight loss when compared to the films with a similar composition and layer number, but produced with unmodified polymers. When BGNPs were incorporated (CatBG50), there was a decrease on the adhesive properties, but still it was much higher than the films with unmodified polymers. Interestingly, the combined effect of the inclusion of catechol and BGNPs resulted in a stiffness of 6.8 and 1.7 times higher than BG50 and Cat50, respectively. The preliminary biological assays indicated that the SaOs-2 cells on the surface of the CTR50, Cat50, and CatBG50 were viable and able to proliferate over a period of 7 days. CatBG50 presented an enhanced cellular behavior, since cells tend to prefer stiffer and less hydrated membranes and the effects of the BGNPs’ ion dissolution on cells metabolic activity. Overall, the use of either catechol-modified polymers or BGNPs by itself did not affect negatively cell viability. So, one may conclude that these LbL films are noncytotoxic, assuring the materials’ biocompatibility.

In conclusion, bioactive LbL freestanding films with enhanced adhesion and mechanical properties were successfully developed through SA-LbL. A synergetic effect on the films’ overall properties was observed upon the combined incorporation of catechol-modified polymers with BGNPs. Such films also presented hydrophilic surfaces, reduced biodegradability, and decreased swelling and promoted higher cellular metabolic activity. These bioactive LbL freestanding films that combine good adhesion with improved mechanical properties could find applications in the biomedical field, such as guided hard tissue regeneration (GTR) membranes.

## Figures and Tables

**Figure 1 molecules-25-00840-f001:**
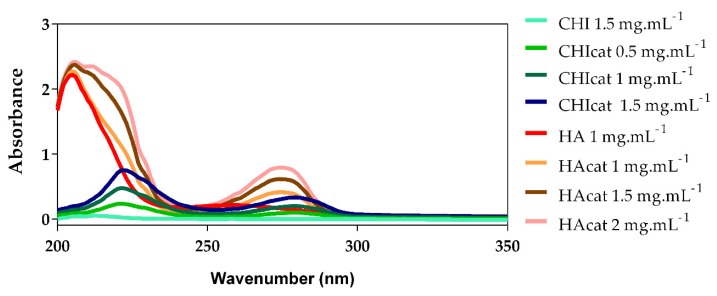
UV-Vis spectra of chitosan (CHI) and hyaluronic acid (HA), catechol-conjugated CHI (CHIcat) and catechol-conjugated HA (HAcat) (λ = 200–350 nm).

**Figure 2 molecules-25-00840-f002:**
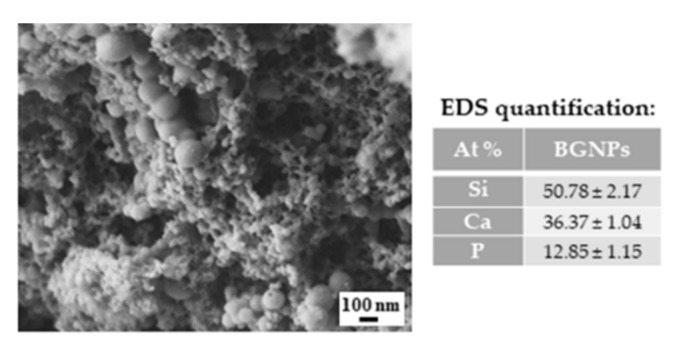
SEM image of the produced ternary BGNPs and EDS quantification (At. %) of the silica (Si), calcium (Ca), and phosphorous (P) presence in the BGNPs.

**Figure 3 molecules-25-00840-f003:**
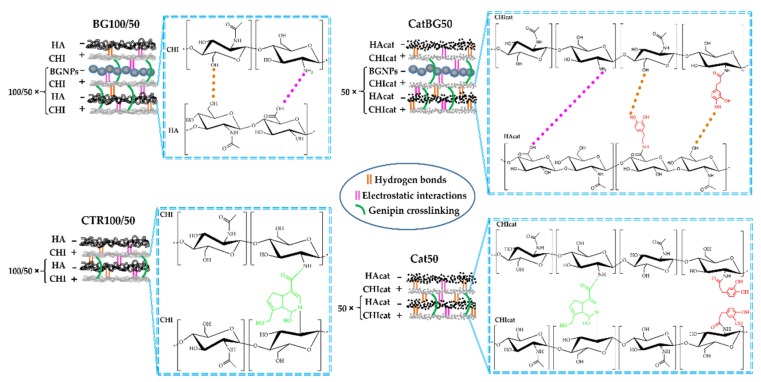
Schematic representation of the distinct freestanding film formulations with the chemical structure of each compound and their possible chemical interactions.

**Figure 4 molecules-25-00840-f004:**
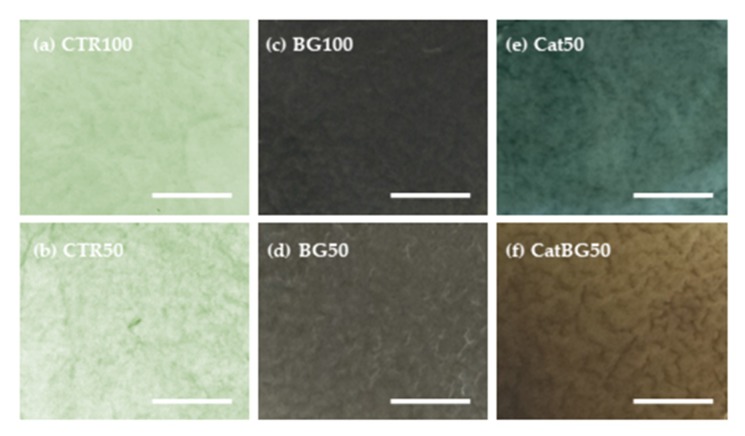
Photographs of the produced LbL freestanding films after cross-linking with genipin (**a**) CTR100, (**b**) CTR50, (**c**) BG100, (**d**) BG50, (**e**) Cat50, and (**f**) CatBG50. Scale bars represent 0.5 cm.

**Figure 5 molecules-25-00840-f005:**
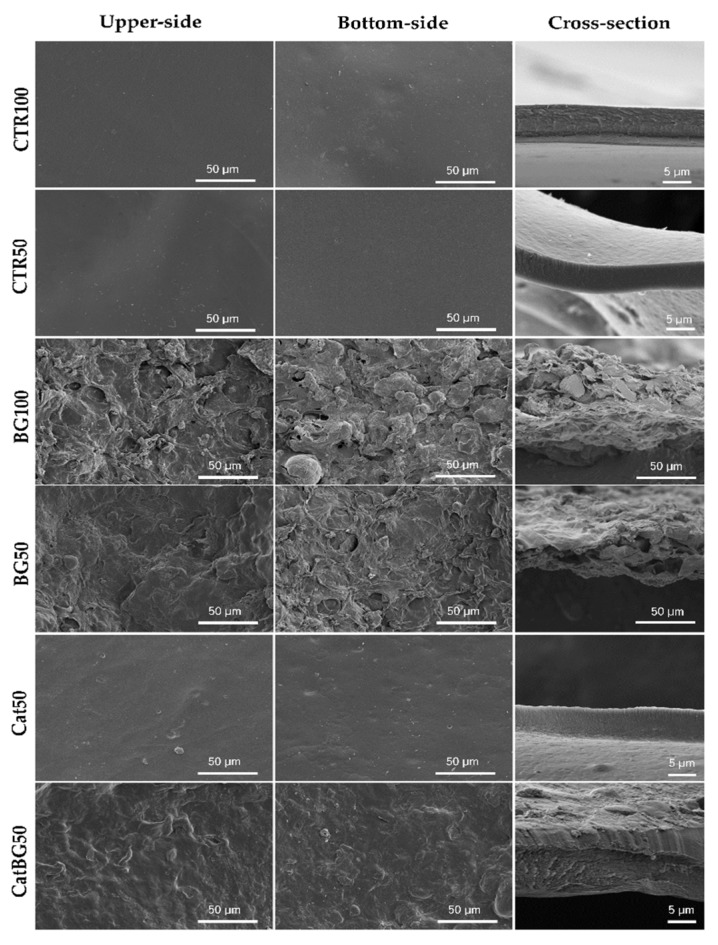
SEM micrographs of cross-linked LbL freestanding films.

**Figure 6 molecules-25-00840-f006:**
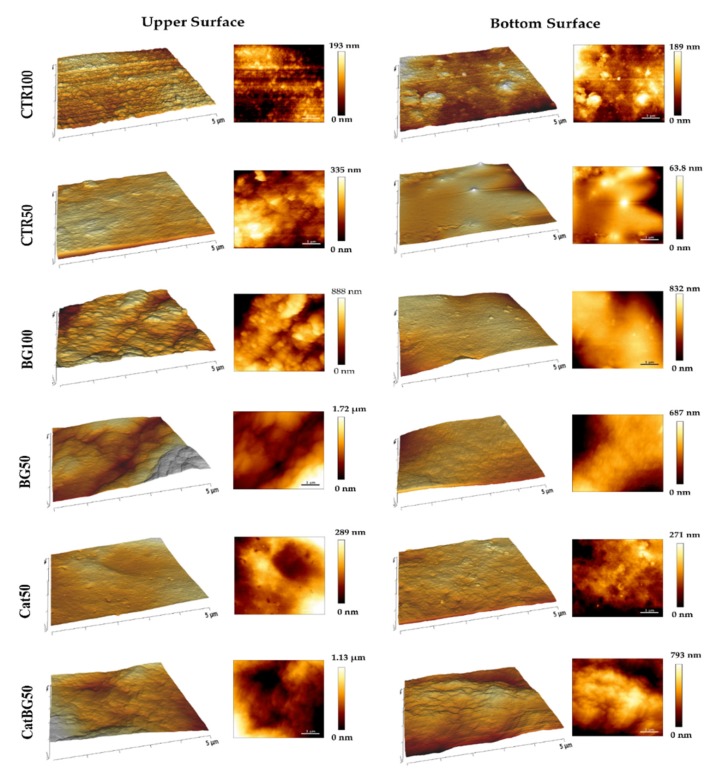
AFM surface images with respective 3D representation of each surface of the LbL films (5 μm × 5 μm).

**Figure 7 molecules-25-00840-f007:**
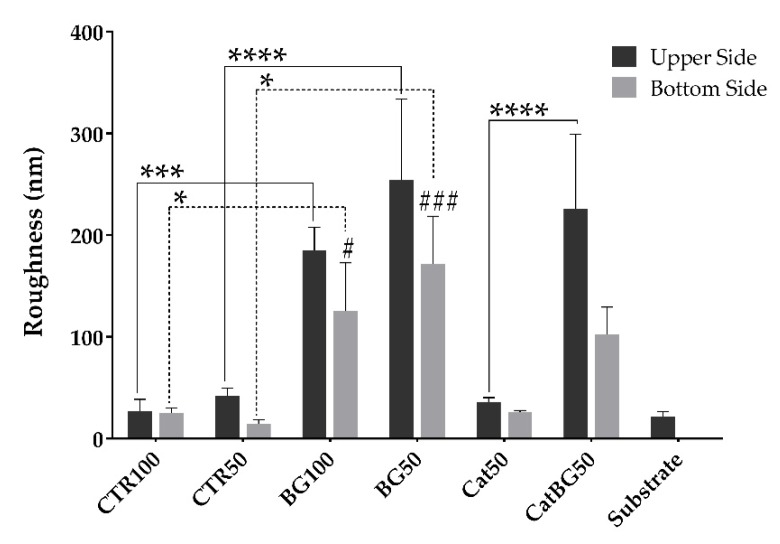
Root mean squared (RMS) roughness of each surface of the produced LbL freestanding films, where the “bottom side” stands for the films’ surface in contact with the substrate during production. Significant differences were found for *p* < 0.05 (*), *p* < 0.01 (**), *p* < 0.001 (***), *p* < 0.0001 (****), between the same surfaces of each film. Significant differences were found for *p* < 0.05 (#), *p* < 0.001 (###) between the corresponding film bottom surfaces and the substrate surface. In straight and dotted line are represented the significant differences between upper surfaces, and between bottom surfaces, respectively.

**Figure 8 molecules-25-00840-f008:**
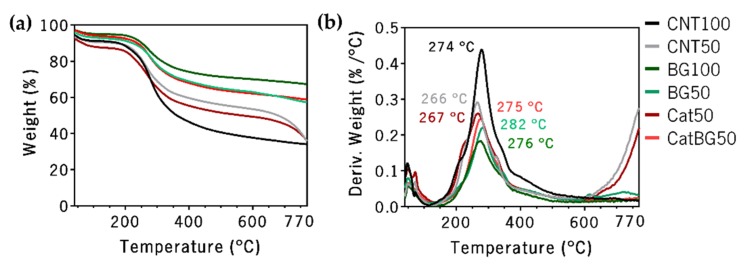
TGA thermograms of (**a**) weight loss and (**b**) derivative of the weight loss (DTGA) of the freestanding LbL films as a function of temperature.

**Figure 9 molecules-25-00840-f009:**
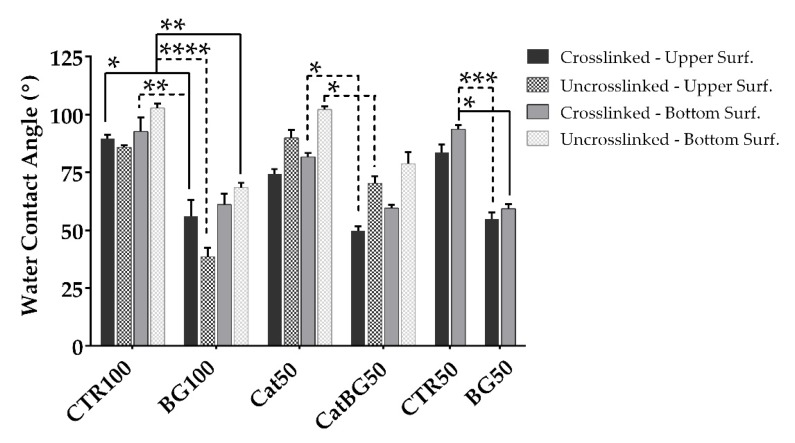
Water contact angle (WCA) measured values of both sides of the films before (except CTR50 and BG50) and after cross-linking with genipin, being the “upper surface” HA/HAcat and the “bottom surface” CHI/CHIcat in all film formulations. Significant differences were stated for *p* < 0.05 (*), *p* < 0.01 (**), *p* < 0.001 (***), and *p* < 0.0001 (****). Straight line represents the significant differences between equivalent surfaces and in dotted lines those between opposite surfaces (i.e., upper-bottom).

**Figure 10 molecules-25-00840-f010:**
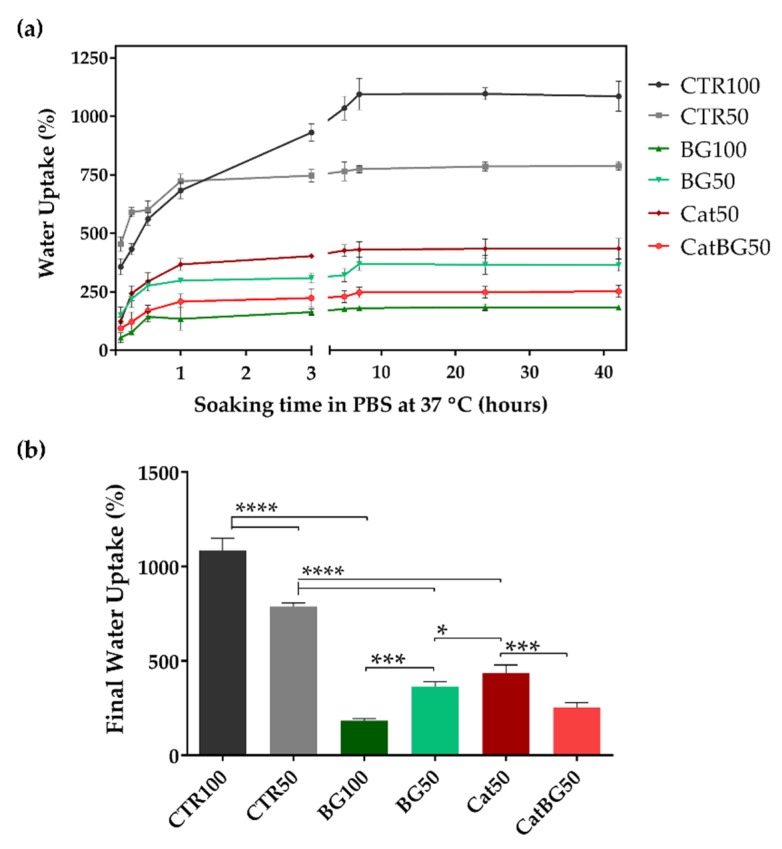
Water uptake ability of the LbL films after immersion in PBS for 42 h at 37 °C. (**a**) Water uptake percentage as a function of time; (**b**) final water uptake percentage. Significant differences were found for *p* < 0.05 (*), *p* < 0.001 (***) and *p* < 0.0001 (****).

**Figure 11 molecules-25-00840-f011:**
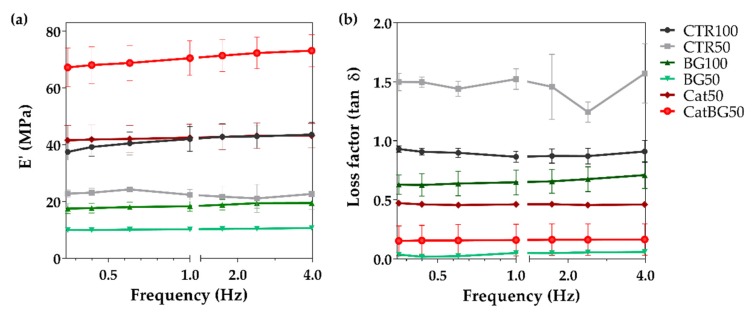
Dynamic mechanical analysis (DMA) measurements: Variation of the storage modulus (**a**) and loss factor (**b**) along a frequency scan ranging from 0.1 to 10 Hz at 37 °C, of the LbL films.

**Figure 12 molecules-25-00840-f012:**
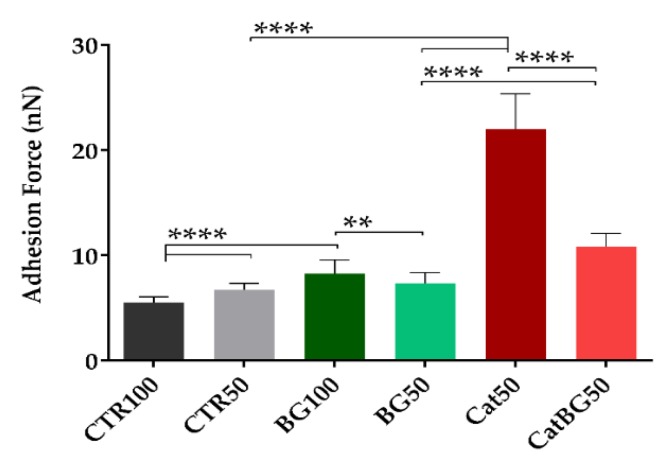
Adhesive force of the LbL films determined by AFM in dry state. Significant differences were found for *p* < 0.01 (**) and *p* < 0.0001 (****).

**Figure 13 molecules-25-00840-f013:**
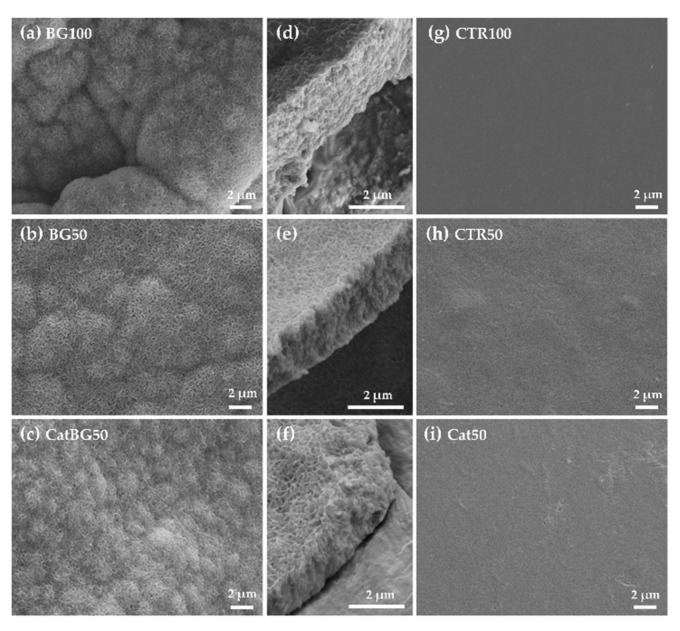
SEM images (**a**), (**b**) and (**c**) of the surface of the bioactive LbL films after immersion in SBF for 14 days, where (**d**), (**e**), and (**f**) present a cross-sectional view of the apatite layer formed onto the films’ surface. (**g**), (**h**) and (**i**) present the surface of the non-bioactive LbL films.

**Figure 14 molecules-25-00840-f014:**
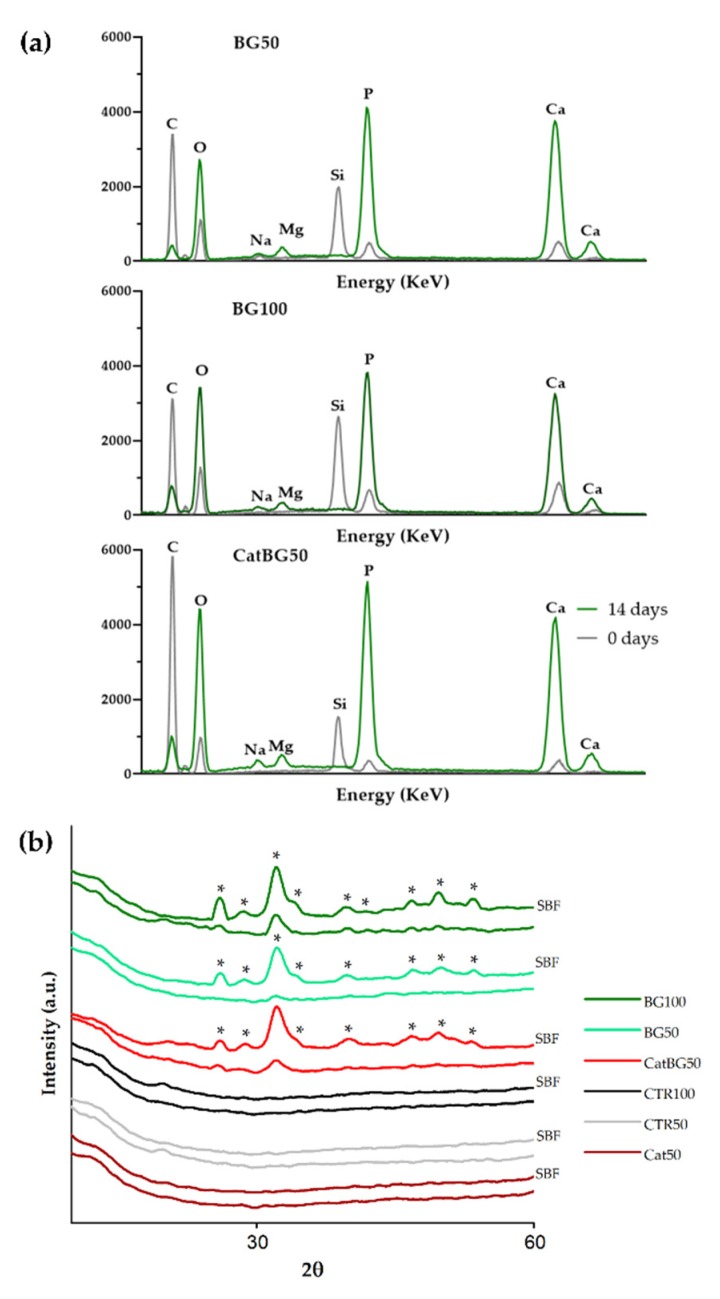
In vitro bioactivity study of the LbL films after immersion in SBF for 14 days: (**a**) EDS and (**b**) XRD spectra. Hydroxyapatite peaks are indicated (*).

**Figure 15 molecules-25-00840-f015:**
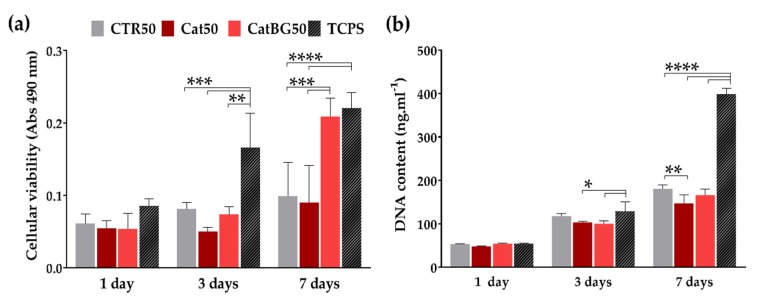
(**a**) Cellular viability analysis using the MTS assay and (**b**) DNA quantification using the PicoGreen assay, SaOs-2 cells cultured on the CTR50, Cat50, and CatBG50 films in D-MEM for 1, 3, and 7 days. Tissue culture polystyrene (TCPS) was used as control. Significant differences were found for *p* < 0.05 (*), *p* < 0.01 (**), *p* < 0.001 (***), and *p* < 0.0001 (****).

**Table 1 molecules-25-00840-t001:** Zeta potential measurements of CHI, CHIcat, HA, HAcat, and bioglass nanoparticles (BGNPs).

Polyelectrolyte	CHI	CHIcat	HA	HAcat	BGNPs
ζ (mV)	23.37 ± 2.22	19.47 ± 1.80	−20.43 ± 1.65	−18.40 ± 1.80	−20.50 ± 0.76

**Table 2 molecules-25-00840-t002:** Multilayered formulations and spin-coating parameters used to assemble spin-coated layer-by-layer (LbL) freestanding films.

Formulations		Spin Coating Parameters	
CTR100	(CHI-HA-CHI-HA)*100-(CHI-HA)	Deposit amount	1.5 mL *
CTR50	(CHI-HA-CHI-HA)*50-(CHI-HA)	Spin speed	2000 rpm
BG100	(CHI-HA-CHI-BGNPs)*100-(CHI-HA)	Spin acceleration	1300 m s^−1^
BG50	(CHI-HA-CHI-BGNPs)*50-(CHI-HA)	Spinning time	10 s
Cat50	(CHIcat-HAcat-CHIcat-HAcat)*50-(CHIcat-HAcat)	Washing spin time	5 s
CatBG50	(CHIcat-HAcat-CHIcat-BGNPs)*50-(CHIcat-HAcat)	Contact time	5–7 s

* 3 mL on the first tetralayer.

**Table 3 molecules-25-00840-t003:** Dry thickness of the LbL films before and after cross-linking with genipin estimated through the SEM images using the ImageJ software (National Institutes of Health, Bethesda, MD, USA).

Formulations	Thickness (µm)	
	Crosslinked	Uncrosslinked
CTR100	5.98 ± 0.06	6.15 ± 0.10
CTR50	5.12 ± 0.05	-
BG100	53.26 ± 5.53	48.56 ± 2.01
BG50	40.68 ± 4.68	-
Cat50	7.58 ± 0.12	3.35 ± 0.21
CatBG50	13.06 ± 0.57	10.5 ± 0.62

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
