# Peer review of "Spin-Coated Polysaccharide-Based Multilayered Freestanding Films with Adhesive and Bioactive Moieties"

_molecules, 2020, doi:10.3390/molecules25040840_

Round 1
Reviewer 1 Report
Authors described an important work in the area of materials chemistry, with a large volume of results. However, the article is very long which leads to lack of focus of the manuscript. In this sense the manuscript must be reviewed and the authors should focus on the main objective of the paper and then part of the data must be presented as supplementary material. Some points are detailed below.
-Introduction section is too long.
-Figure quality (in general) is to low.
-Figure 1 and A1: Authors must clear discuss their UV-visible spectra. Those curves (blue and dark greens) present a large difference in baseline, full spectra must be presented as supporting information. It seems that there was some particles in the solution. In addition, Figure A1 quality must be improved, the value of integrations can barely be observed. Moreover, chemical shift must be presented in a table and hydrogens must be assigned.
-Authors should change M (molar) for mol/L.
-Some sections are not connected to each other. (e.g. 2.1 and 2.2).
-Captions must be revised: Figure 2 must be changed, side table must be presented in captions. Figure 8(b) is not mentioned in the caption. In addition, due to its low quality Figure 8(b) must be removed.
-Authors should avoid large number of self-citation (more than 10%).
Author Response
Please see the attachment with our responses.

Reviewer 2 Report
Alves and coworkers reported a spin-coated polysaccharide-based film for adhesion and bioactivity. Several points should be addressed before further consideration.
The authors did not clearly emphasize the importance of the research, especially missing how it connects with biological applications. Instead, the authors refers the applications as "in wound healing, surgical tissue adhesives, drug delivery depots and tissue engineering scaffolds, among others". Honestly, these are vague claims. A general question for the study as a whole: how will spin-coating be translated to tissue engineering? Will they form a film that is easily peeled off from the production and have enough stability/biocompatibility? The representation of the manuscript overall needs to add illustrations on the chemistry, e.g. the chemical structure of each component, the reactions involving these materials, the crosslinking chemistry of genipin. The figure quality is highly suggested to be improved, such as Figure 3, 5, and 7. What is the basic information that the authors try to convey from Figure 10? If no difference is observed with or without enzymes in PBS, the figure after all is not significantly important to the study.Author Response
Please see the attachment with our responses.

Round 2
Reviewer 1 Report
Authors have improved their manuscript introduction section. In addition, they have also removed large number of self-citation. However, they have not enough improved their results and discussion section.
-Figure 1: y axis (absorbance does not present unit), then remove (a.u). In addition, authors should avoid to use electronic measurements for samples at high concentrantion, leading to values of absorbance above 2. These data are out of Beer Lambert linearity. Authors should present their data of diluted solution.
-Authors must clearly state how the table A1 confirms the chemical modification, since only "Most relevant chemical shifts in 1H-NMR spectra of CHI, CHIcat and HAcat" are presented. Authors should include 13C to compare the chemical modification.
-Zeta Potential Measurements were not conclusive, since values were very similar, this could be and indicative of low degree of substitution or even no substitution.
-Authors must review their section numbers: "2.4. Production of the LbL freestanding films" and then, "2.4.Films morphological and topographic characterization"
-For some results (e.g.: Zeta Potential Measurements and Thermogravimetric analysis), the authors do not make clear the purpose of using the technique.
-in general, the authors must clearly present the objective of all the techniques used for the purpose of the present work. Avoiding the use of a large volume of data that are not correlated.
Author Response
Please see our responses in attachment.

Reviewer 2 Report
Please work on the language of this manuscript. It needs to be improved.
Author Response
Please see our response in attachment.
